# Deposition and Characterization of RP-ALD SiO_2_ Thin Films with Different Oxygen Plasma Powers

**DOI:** 10.3390/nano11051173

**Published:** 2021-04-29

**Authors:** Xiao-Ying Zhang, Yue Yang, Zhi-Xuan Zhang, Xin-Peng Geng, Chia-Hsun Hsu, Wan-Yu Wu, Shui-Yang Lien, Wen-Zhang Zhu

**Affiliations:** 1School of Opto-Electronic and Communication Engineering, Xiamen University of Technology, Xiamen 361024, China; xyzhang@xmut.edu.cn (X.-Y.Z.); 1922031005@stu.xmut.edu.cn (Y.Y.); 1922031023@stu.xmut.edu.cn (Z.-X.Z.); 1922031048@stu.xmut.edu.cn (X.-P.G.); chhsu@xmut.edu.cn (C.-H.H.); wzzhu@xmut.edu.cn (W.-Z.Z.); 2Fujian Key Laboratory of Optoelectronic Technology and Devices, Xiamen University of Technology, Xiamen 361024, China; 3Department of Biomedical Engineering, Da-Yeh University, Chung Hua 51591, Taiwan; wywu@mail.dyu.edu.tw

**Keywords:** SiO_2_ thin film, oxygen plasma power, atomic layer deposition

## Abstract

In this study, silicon oxide (SiO_2_) films were deposited by remote plasma atomic layer deposition with Bis(diethylamino)silane (BDEAS) and an oxygen/argon mixture as the precursors. Oxygen plasma powers play a key role in the quality of SiO_2_ films. Post-annealing was performed in the air at different temperatures for 1 h. The effects of oxygen plasma powers from 1000 W to 3000 W on the properties of the SiO_2_ thin films were investigated. The experimental results demonstrated that the SiO_2_ thin film growth per cycle was greatly affected by the O_2_ plasma power. Atomic force microscope (AFM) and conductive AFM tests show that the surface of the SiO_2_ thin films, with different O_2_ plasma powers, is relatively smooth and the films all present favorable insulation properties. The water contact angle (WCA) of the SiO_2_ thin film deposited at the power of 1500 W is higher than that of other WCAs of SiO_2_ films deposited at other plasma powers, indicating that it is less hydrophilic. This phenomenon is more likely to be associated with a smaller bonding energy, which is consistent with the result obtained by Fourier transformation infrared spectroscopy. In addition, the influence of post-annealing temperature on the quality of the SiO_2_ thin films was also investigated. As the annealing temperature increases, the SiO_2_ thin film becomes denser, leading to a higher refractive index and a lower etch rate.

## 1. Introduction

Silicon dioxide (SiO_2_) is a versatile material that is used for many applications. SiO_2_ and more generally ultra-thin oxide films have been extensively described as good components for modern nano technologies such as dielectric materials in silicon microelectronic devices, anticorrosion films, or non-exhaustive applications of nanoscale films in catalysis [1]. SiO_2_ films can be prepared by many methods, such as chemical vapor deposition (CVD) [2,3,4], thermal oxidation [5,6,7], and atomic layer deposition (ALD) [8,9,10,11,12,13,14,15]. Among these preparation methods, the ALD method is highlighted because the film growth can be precisely controlled based on self-limiting and sequential surface reactions. ALD can be used to prepare various thin films, such as alumina, graphene-augmented alumina [16,17,18,19], Hafnium dioxide, SiO_2_, and so on. Many studies on the deposition of SiO_2_ films through ALD have been conducted. In 2018, M. Ziegler et al. [11] prepared 3D nanostructures using silica with tri(dimethylamino)silane and oxygen plasma. During the same year, Yoo-Jin Choi et al. [12] reported that SiO_2_ deposited by ALD functions as gate dielectric thin films. In 2019, Wen Zhou et al. [13] investigated ALD SiO_2_ for the channel isolation of colloidal quantum dot phototransistors. D. Arl et al. [1] reported less contaminated and porous SiO_2_ films grown by ALD at room temperature. Marc, J. M. Merkx et al. [15] studied the removal and reapplication of the inhibitor molecules during the area-selective ALD of SiO_2_. There are various ALD process modes. Remote plasma atomic layer deposition (RP-ALD) is a widely used mode. The RP-ALD possesses the high reactivity of many plasma species; it can thus reduce the deposition temperatures without degrading the prepared film quality. After comparing many investigations, Bis(diethylamino)silane (BDEAS) is the usual precursor for depositing SiO_2_ due to its chemical stability. Argon (Ar) and oxygen (O_2_) are the most commonly used gases that can effectively generate plasma because of their stability and unique electronegativity [20,21], respectively. Ar and O_2_ molecules are activated into excited atoms. The functional groups of the precursor are then oxidized by the excited atoms. However, systematic investigations on the oxygen plasma power for the property of SiO_2_ films are rare.

SiO_2_ films were grown by RP-ALD with different oxygen plasma powers. The variation of the oxygen plasma power on the effects of the electrical, optical, chemical, structural, and morphological properties of the films was investigated. Furthermore, the annealing temperature is another parameter that affected the quality of the thin films prepared by the RP-ALD. Therefore, the relationship between the property changes of the SiO_2_ thin films and the variation in the annealing temperature is also discussed.

## 2. Experimental Section

Four-inch (100) oriented p-type Czochralski polished Si wafers were used as substrates. The Si wafer had a resistivity of about 50 Ω-cm and a thickness of about 450 μm. Prior to the deposition of the SiO_2_ thin films, all wafers were cleaned using deionized water for 10 s, a diluted hydrofluoric (HF) acid solution (2%) for 1 min, and deionized water for 10 s. After cleansing, all Si wafers were dried using pure nitrogen (N_2_) gas with a purity of 99.99%, then transferred onto the substrate holder of a commercial RP-ALD system (Picosun R-200, Espoo, Finland). The RP-ALD system had six source channels. One of the channels was connected to the BDEAS bubbler, and the rest of the channels were connected to N_2_. The plasma was produced in a microwave cavity using inductive coupling of radio frequency (RF) power with the mixture of Ar and O_2_ gases. These two gases both have a high purity of 99.999%. The SiO_2_ thin films were grown using BDEAS (Aimou Yuan, Nanjing, China) and remote O_2_ plasma as the precursors with the carrier gas N_2_. The BDEAS precursor foreline was heated to 45 °C in order to avoid any precursor condensation. The base pressure of the system was 100 Pa. One BDEAS pulse and one O_2_ plasma pulse were 1.6 s and 8 s respectively. The N_2_ purge times for BDEAS and O_2_ were both 5 s. The substrate holder was heated to 250 °C and kept at the same temperature in the deposition process. O_2_ plasma at different powers (1000, 1500, 2000, 2500, and 3000 W) was used. Table 1 lists the detailed experiment parameters of the SiO_2_ thin films. The SiO_2_ thin films with a plasma of 2000 W were separately prepared and annealed at 300–850 °C in the air for 1 h to investigate their densification. Various SiO_2_ thin films were obtained by chemical etching using a diluted HF acid (2% in deionized H_2_O), followed by a deionized water rinse.

The O_2_ plasma power emission spectra were assessed by an optical emission spectrometer (OES). The thickness and refractive index of the SiO_2_ thin films deposited on the silicon wafers were measured by a spectroscopic ellipsometer (SENTECH SE 800 DUV, Berlin, Germany) using an air roughness model, which consisted of an “air, SiO_2_, silicon” three-layer structure. The refractive index was extracted from the spectroscopic ellipsometer measurements assuming a Cauchy dispersion model. AFM (Bruker, Karlsruhe, Germany) measurements were performed to investigate the surface morphology of the SiO_2_ thin films. The AFM images shown in this study are 5 μm × 5 μm scans with a resolution of 256 points × 256 lines. All the AFM evaluations to calculate roughness were conducted from the central areas of the samples. Conductive atomic force microscopy (CAFM) was also measured in the air to analyze the insulation properties of the SiO_2_ thin films. The bonding configuration of the SiO_2_ thin films was examined by Fourier transformation infrared (FTIR) spectroscopy (Bruker, Karlsruhe, Germany). The water contact angle (WCA) and the surface free energy (SFE) of the SiO_2_ thin films were also evaluated by a contact angle analyzer (Dongguan Shengding Precision Instrument Co., Ltd., Dongguan, China).

## 3. Results and Discussion

The image of the visual observation of glow discharge at the plasma power of 2000 W is presented for reference in Figure 1. The O_2_ plasma pulse was produced by a dielectric tube. The distance between the sensing point and the substrate was 430 mm. According to a research study carried out by the Zhen Zhu group, there is a recognizable effect of the oxygen plasma power on the SiO_2_ thin films’ impurity levels [22]. The OES method has also proven to be useful as a tool for the monitoring of film deposition by PECVD [23]. Figure 2a shows the OES spectra of SiO_2_ thin films prepared with an O_2_/Ar mixture plasma power of 1000, 1500, 2000, 2500, and 3000 W. The main features of the OES spectra correspond to the emission of Ar ions and O radicals. Ar ions and O radicals’ emission spectra were assigned according to the signals from pure Ar and O_2_ plasma [24]. The emission spectra are caused by the transition of the electrons. When the plasma concentration is lower, such as between 1000 and 1500 W, the intensity of the Ar ions and O radicals is low in the wavelength region of 680 to 800 nm, indicating that the ionization of the gases is low. Although the intensity of Ar ions and O radicals is enhanced at the plasma power of 1500 W, there is not enough energy to fully oxidize the BDEAS. When the plasma power is increased to 2000 W, the intensity of the O radicals increases significantly and surpasses the intensity of Ar ions. The spectra are similar when the powers are between 2000 and 3000 W. Figure 2b shows the summed intensity of Ar ions and O radicals at various plasma powers. When the plasma power is increased, both the summed intensities of Ar ions and O radicals are increased and then tend to saturate. When the plasma power is at 1000 and 1500 W, the summed intensity of Ar ions and O radicals is considerably low, suggesting that the ionization of the gases is quite low. When the plasma power is above 2000 W, the summed intensity of the O radicals increases sharply and begins to saturate. The summed intensity of O radicals exceeds the summed intensity of Ar ions, indicating that the ionization of the gases is sufficient. However, the summed intensity of Ar ions continues to increase, which may increase the plasma bombardment etching effect, resulting in the degradation of the SiO_2_ film quality.

Figure 3a reveals a linear increase in the thickness of the SiO_2_ thin films as a result of different cycles for five different plasma powers. As shown in Figure 3a, the SiO_2_ thin films’ thickness increases as the ALD cycle increases from 100 to 1000. The linear relationship between the SiO_2_ thin films’ thickness and the cycles has confirmed that the SiO_2_ thin films were prepared in a self-limiting manner [25]. Figure 3b shows the relationship between O_2_ plasma power and the deposition rate of the SiO_2_ thin films. The deposition rate was determined using the SiO_2_ thin film thickness divided by the total number of ALD cycles. The growth rate is about 0.1 nm/cycle at a plasma power of 1000 W. The growth rate sharply plunged to a minimum of about 0.04 nm/cycle when the plasma power increased to 1500 W. When the plasma further increased, the growth rate rose again and became saturated. The highest deposition rate at a plasma power of 1000 W may arise from the deficient energy of oxygen radicals, which grow loose SiO_2_ molecules or SiO. According to the OES measurement, the intensity of O radicals is relatively low. Therefore, when the SiO_2_ thin films are deposited, the surface reactions may deviate from the ideal ALD process, giving rise to an abnormally high deposition rate at 1000 W. Under the circumstances, the SiO_2_ thin films were prepared with a considerably loose structure instead of a typical ALD film structure. When the plasma power goes up to 1500 W, the BDEAS reacts with the hydroxyl groups which are on the substrate surface, and the excess BDEAS molecules and byproducts are purged by N_2_. Then, greater amounts of oxygen radicals oxidize the BDEAS absorbed on the substrate surface to form an OH terminated surface. However, the amount of oxygen radicals is still insufficient to form a dense SiO_2_ film. When the plasma power is higher than 2000 W, adequate oxygen radicals take part in the surface oxidation reaction, leading to a higher deposition rate. When the plasma power is changed from 2000 W to 3000 W, the Ar ions boost the bombardment etching effect, as more Ar ions are excited. This is confirmed by the OES measurement. The deposited rate is gained by the trade-off between the plasma bombardment etching effect and the full oxidation. It is interesting to note that this variation is also found in the deposition of tin oxide prepared by the same ALD system in our lab [26].

Figure 4a shows the measured wavelength-dependent refractive index spectra for the RP-ALD SiO_2_ films with various plasma powers. As can be seen from the graph, the refractive index fluctuates between 1.46 and 1.47, which is comparable to the value of 1.465 for the thermally grown SiO_2_ film prepared at 1000 °C in dry O_2_ [6]. The values at plasma power of 1000 and 1500 W are higher. This could be related to the stoichiometric ratio of the SiO_2_ thin films. When depositing the SiO_2_ thin films at a lower power of 1000 and 1500 W, the O radicals were not enough to oxidize the BDEAS absorbed on the Si substrate, resulting in oxygen deficit. Thus, some SiO with a higher refractive index of about 1.8~1.9 [27] or SiO_x_ (1 < x < 2) was formed at the interface. As the O_2_ plasma power was increased to 2000, 2500, and 3000 W, more O radicals were produced with sufficient radicals to form SiO_2_ films. Therefore, the refractive indexes at a higher O_2_ plasma power show little change. Figure 3b displays the refractive index spectra of the RP-ALD SiO_2_ thin films at various annealing temperatures. The films were all deposited at a plasma power of 2000 W. From this diagram, the refractive index of the annealing samples are all higher than the refractive index of the non-annealing samples. When the annealing temperature increase from 400 to 850 °C, the refractive index of the annealing SiO_2_ thin films continuously increases. The noticeable increase in the refractive index may be a reflection of the change in film density, as they are closely related. When the SiO_2_ thin films were annealed from 400 to 850 °C, they became compact, giving rise to a higher refractive index.

The wet etch rate is a key parameter to determine the densification of the annealing SiO_2_ thin films. In this study, the wet etch rate was obtained by immersing the SiO_2_ thin films in a diluted 2% HF acid solution. Figure 5a shows the etched thickness of the SiO_2_ thin films deposited at 2000 W versus the etching time. The etched SiO_2_ thickness gradient was measured using a spectroscopic ellipsometer. The etch rate of the samples in the HF acid solution is determined by the relationship between the SiO_2_ thickness and the etching time gradient. The etching rate is denoted in nanometer per second. As shown in the figure, the thickness of as deposited SiO_2_ thin film rapidly decreases when the etching time increases, indicating that the SiO_2_ thin film has insufficient densification. In addition, the etch rate decreases as the annealing temperature increases. The densification of SiO_2_ films is improved by the annealing process. Figure 5b depicts a variation in the etch rate of the SiO_2_ thin films with 70 s etching when the temperature annealed from 300 to 800 °C. It is noted that the etch rate decreases sharply and then decreases slowly when the annealing temperature gradually increases. The etch rate at an annealing temperature of 200 °C is about 1.6 nm/s, while it decreases to about 0.3 nm/s when the annealing temperature is 800 °C. The SiO_2_ thin films were easily decomposed in a diluted HF acid solution because they were not fully densified when their annealing temperatures were lower [28]. This result is also supported by the change in refractive index, because it commonly corresponds to the densification of the SiO_2_ thin films. When the annealing temperature increases, the SiO_2_ thin film becomes denser, leading to a lower etch rate. The SiO_2_ films’ etching mechanism in HF acid solution can refer D. Martin Knotter’s study. According to a research study carried out by D. Martin Knotter [29], the dissolution of vitreous silicon dioxide in HF-based solutions depends on the number of silicon atoms bonded to four bridging oxygen atoms, the pH, and the concentration of HF_2_^−^.

The morphological evolution of the SiO_2_ thin films prepared at various O_2_ plasma powers was confirmed by AFM. Figure 6 shows the AFM images for the SiO_2_ thin films deposited at different O_2_ plasma powers: (a) 1000 W, (b) 1500 W, (c) 2000 W, (d) 2500 W, and (e) 3000 W. Since plasma may change surface roughness, the surface morphology of the SiO_2_ thin films deposited at various plasma powers was studied. The average surface roughness (Ra) values are also plotted in Figure 6f to indicate surface roughness. As can be seen in the figure, the Ra values, which vary from 0.4 nm to 0.5 nm with the plasma power from 1000 W to 3000 W, show little change. This result indicates that all of the surfaces of the RP-ALD SiO_2_ films are relatively smooth. Although the SiO_2_ thin films deposited by a higher plasma power of 2500 or 3000 W undergo Ar ions bombardment, they still have low Ra roughness values. It is worth noting that when the O_2_ plasma is 3000 W, more needle tips emerge on the image, possibly due to the stronger Ar ions bombardment.

Scanning probe microscopes are a powerful tool for investigating the electrical quality of dielectrics [30,31]. CAFM has widely been used to measure the electrical conduction of dielectrics. In our study, CAFM measurements were performed in the air and the current maps were obtained by using a constant voltage to the tip with the substrate grounded. The effect of the plasma power on the electrical quality of the SiO_2_ thin films was investigated on the current images of the thin films measured at 3.0 V on the same regions as shown in Figure 6. Figure 7 shows the current maps of the SiO_2_ thin films with various plasma powers. From the images, it can be seen that all the currents of the SiO_2_ thin films are relatively small, with values on the order of femtoamperes (fA), suggesting that all the samples have favorable insulation properties. When the plasma power is 1000 W, most of the currents of the SiO_2_ thin films are approximately 80 fA due to the considerably loose structure of the film. When the plasma power increases to 1500 W, most of the currents of the SiO_2_ thin films are between 75 and 40 fA. It is important to note that most of the currents of the thin films with a plasma power of 2000 W are about 20 fA, which is the lowest value, when compared with other samples. When the plasma power further increases to 2500 W and 3000 W, most of the current values of the thin films are around 90 fA, which may be induced by the ions’ bombardment effect, leading to a reduction in the SiO_2_ thin film quality.

The chemical composition of the SiO_2_ thin films was measured by FTIR. Figure 8 displays a typical FTIR spectrum of the SiO_2_ thin films prepared at different plasma powers. Peaks in the range of 1100–1000 cm^−1^ could be attributed to the absorption of Si-O vibrations. The strong absorption at about 1067 cm^−1^ indicates the presence of Si-O-Si bonds in the SiO_2_ films. These features are consistent with Si-O vibrational features that have been observed previously [9]. Furthermore, the absorbance strength of the thin film prepared at 1500 W is less than that of other films. The relatively weak absorbance from the Si-O-Si combination bands at a plasma power of 1500 W is primarily caused by the lower SFE.

The wetting characteristic of SiO_2_ films has many applications. Figure 9a shows the WCA of the SiO_2_ thin films with various plasma powers and different annealing temperatures. The WCA is known as the angle at which a water-vapor interface meets the solid surface. The WCA can be used as a measurement to display wetting phenomena. On a solid surface, a drop of water may remain a spherical drop or spread to cover the solid surface. When the WCA is less than 90°, wetting occurs. When the WCA is greater than 90°, the water does not wet the surface. In this study, the volume of liquid dropped to the SiO_2_ film surface was 2 μL every time. As can be seen from the figure, the WCAs vary between 20 and 57°, which is indicative of a hydrophilic surface. The hydrophilic nature of the SiO_2_ surface is due to the hydrogen bonding between silanol groups and water molecules [32]. On the surface of the as deposited SiO_2_ thin films, the WCAs show minor changes, as the change of the O_2_ plasma power. However, the WCA of the as deposited SiO_2_ thin film at the power of 1500 W is higher than that of other WCAs of as deposited SiO_2_ films prepared by other plasma powers, indicating that it is less hydrophilic. This phenomenon is more likely to be associated with the smaller bonding energy, which is in accordance with the result obtained by FTIR. According to findings by Murat Barisik and Beskok, the WCA varies with the variation of the interaction intensity of Si-O [33]. The change of the WCAs for the SiO_2_ thin films annealing at different temperatures is shown in Figure 9b. The samples were all prepared with a plasma power of 2000 W. The shapes of the water droplets created on the SiO_2_ thin films are also included in the figure. It is interesting to note that the higher the annealing temperature, the greater the WCA. When the SiO_2_ thin films were annealed, the WCAs of the SiO_2_ thin films sharply increased from 22.8° at 400 °C to 53.5° at 850 °C. The WCA of the thin film annealing at 850 °C is comparable to the value of 50° studied by Eun-Kyeong Kim [34], which was prepared by thermal annealing Si wafer in an oxidizing atmosphere. To further study the change in the surface state of the SiO_2_ thin films with various plasma powers, the SFE was also measured. Figure 10 shows the variation in SFE for five different plasma powers. The SFE was calculated by the Fowkes method without considering the polar force component [35]:(1)$$\gamma_{L}\left(1+\cos\theta \right)=2\sqrt{\gamma_{S}^{d}\gamma_{L}^{d}},$$
where θ is the WCA between solid and liquid, $\gamma_{L}$ is the surface tension of liquid, and $\gamma_{S}^{d}$ and $\gamma_{L}^{d}$ refer to the dispersion force component of SFE for the solid and liquid. Fowkes method is based on the Yong equation. It is derived for a droplet on an ideal surface that is homogeneous, rigid, and smooth. Corresponding to the WCA, the SFE is lower as the WCA is higher. The result is in accordance with the FTIR and the WCA analyses in Figure 8 and Figure 9, respectively.

## 4. Conclusions

In this study, SiO_2_ thin films were prepared by remote plasma atomic layer deposition. The effects of the oxygen plasma power and annealing temperature on the SiO_2_ properties were investigated. The oxygen plasma power plays a key role in fabricating the SiO_2_ thin films, as they affect the components of the Ar ions and O radicals. The plasma power of 2000 W is the optimum power for preparing high-quality SiO_2_ thin films, because at this power adequate oxygen radicals take part in the surface oxidation reaction with less ion bombardment. AFM and CAFM tests show that all the SiO_2_ thin films are relatively smooth and have insulating properties. When the annealing temperature of a SiO_2_ film is increased, the refractive index increases while the etch rate decreases due to the density of the films. The variation in the bonding energy gives rise to the change in the hydrophilic properties. Therefore, the SiO_2_ thin films surface becomes hydrophobic and the surface free energy decreases as the post-annealing temperature increases. This research achievement is greatly beneficial to the application of RP-ALD SiO_2_ thin films in future semiconductor-related industries.

## Figures and Tables

**Figure 1 nanomaterials-11-01173-f001:**
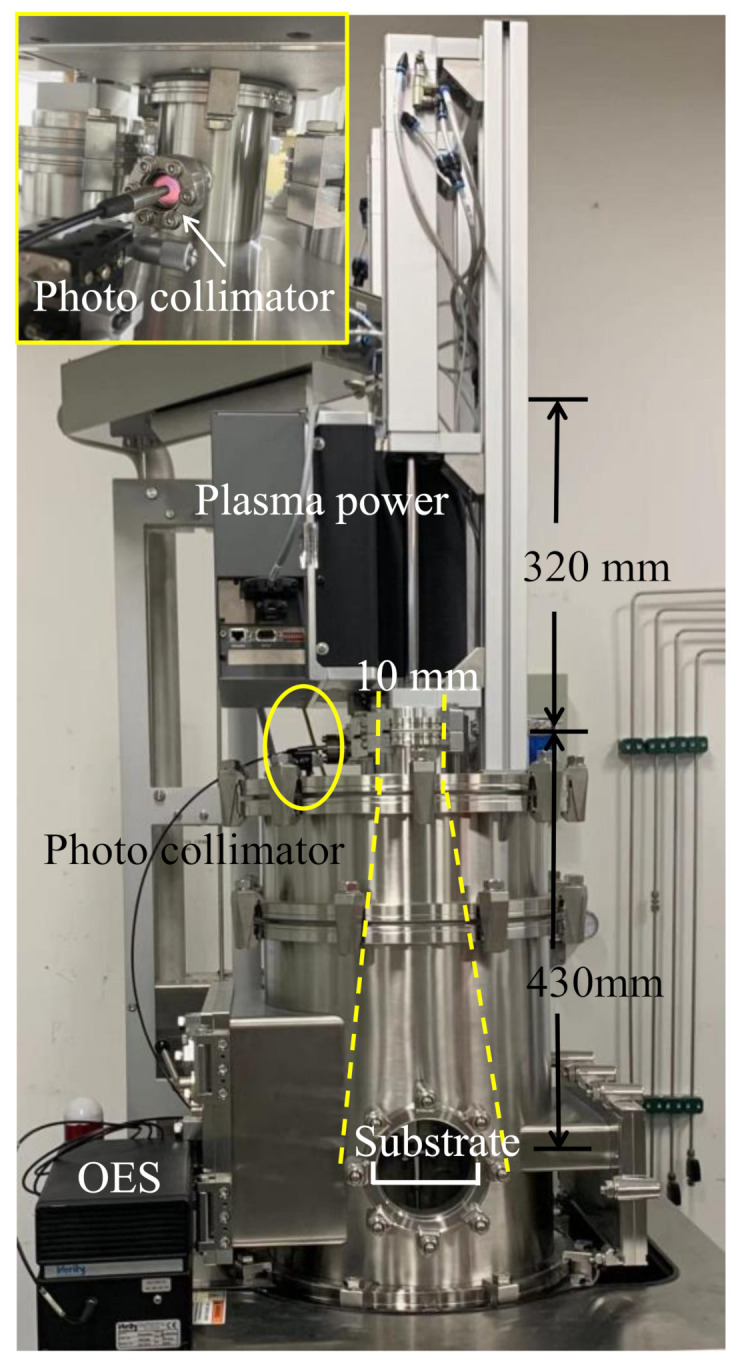
Outline image of the ALD system and visual observation of the glow discharge at 2000 W.

**Figure 2 nanomaterials-11-01173-f002:**
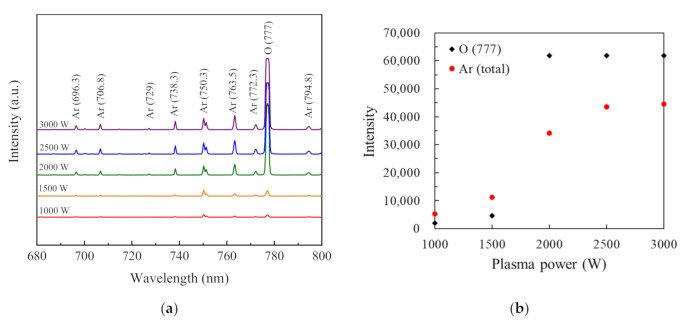
(**a**) OES spectra for SiO_2_ thin films prepared with O_2_/Ar mixture plasma at different plasma powers; (**b**) summed intensity for Ar ions and O radical spectra at various plasma powers.

**Figure 3 nanomaterials-11-01173-f003:**
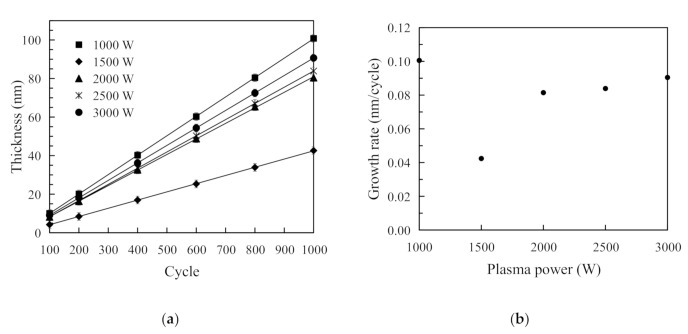
(**a**) Increase in the thickness of SiO_2_ thin films with the number of RP-ALD cycles; (**b**) deposition rate of the RP-ALD SiO_2_ thin films prepared at different plasma powers.

**Figure 4 nanomaterials-11-01173-f004:**
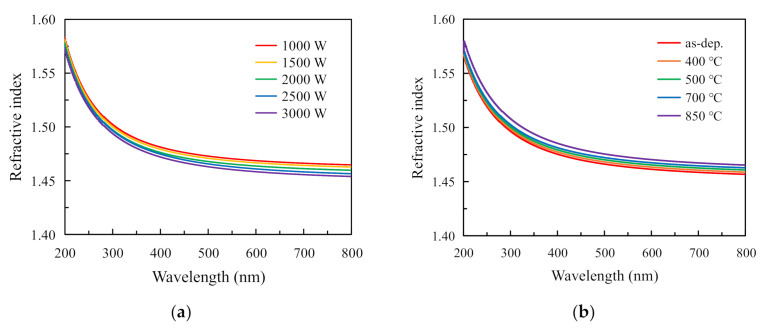
Refractive index spectra for the RP-ALD SiO_2_ thin films (**a**) with various plasma powers; (**b**) with and without different annealing temperatures.

**Figure 5 nanomaterials-11-01173-f005:**
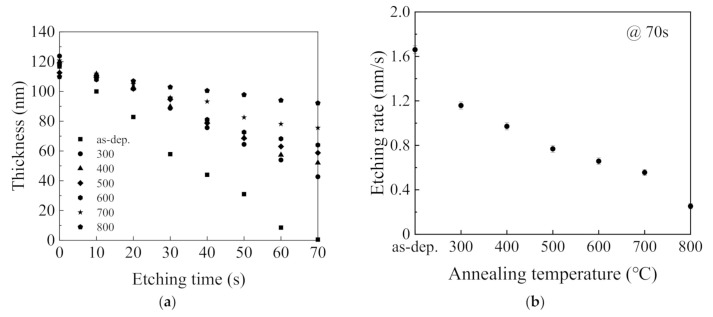
(**a**) Thickness of the RP-ALD SiO_2_ thin films at various annealing temperatures after immersing in diluted HF acid; (**b**) etching rate of the RP-ALD SiO_2_ thin films during the 70 s.

**Figure 6 nanomaterials-11-01173-f006:**
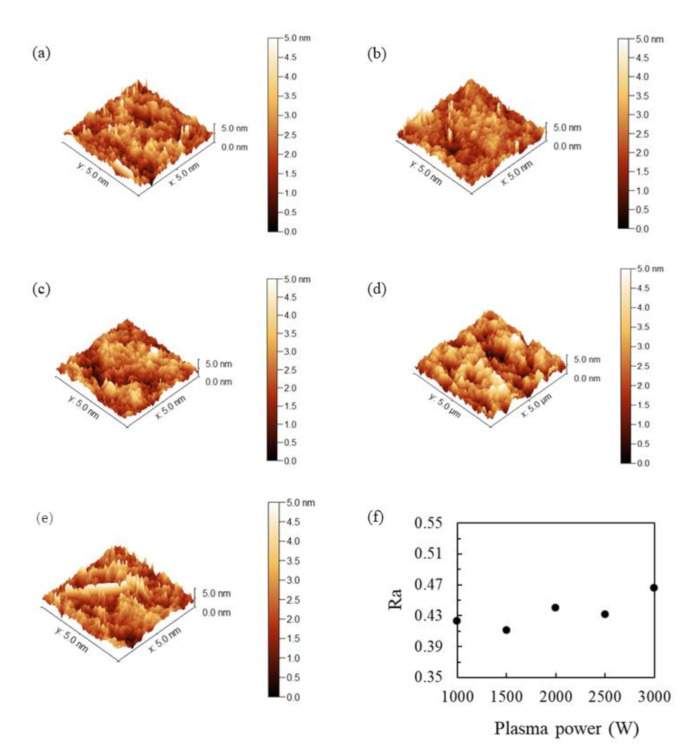
AFM images of the SiO_2_ thin films with plasma power of (**a**) 1000 W, (**b**) 1500 W, (**c**) 2000 W, (**d**) 2500 W, and (**e**) 3000 W. (**f**) Ra values as a function of various plasma powers.

**Figure 7 nanomaterials-11-01173-f007:**
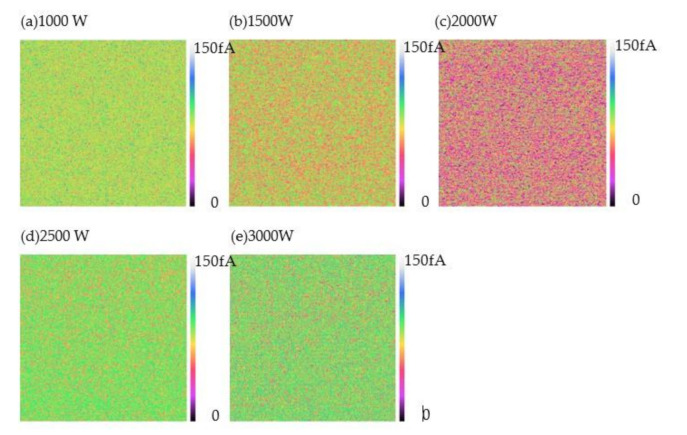
CAFM images of the RP-ALD SiO_2_ thin films with different powers (**a**) 1000 W, (**b**) 1500 W, (**c**) 2000 W, (**d**) 2500 W and (**e**) 3000 W.

**Figure 8 nanomaterials-11-01173-f008:**
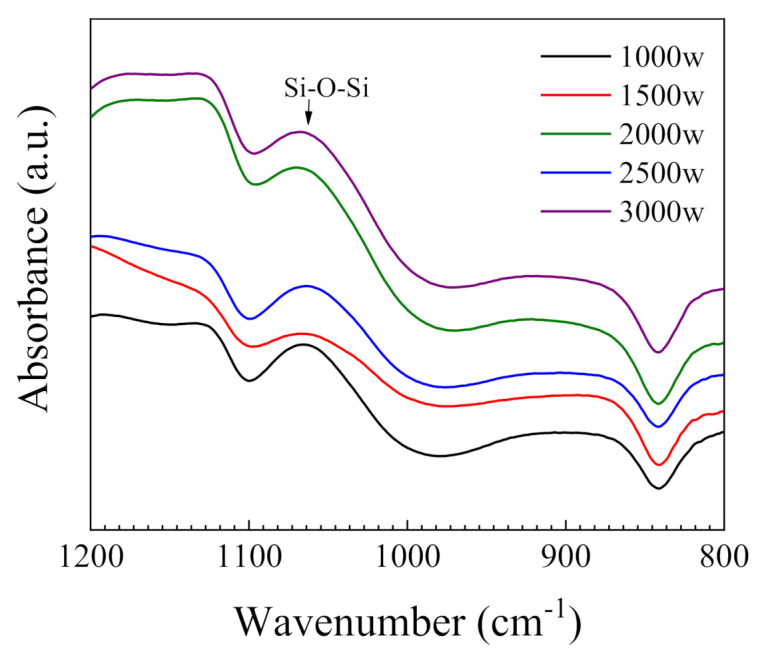
FTIR spectra of the RP-ALD SiO_2_ films with various plasma powers.

**Figure 9 nanomaterials-11-01173-f009:**
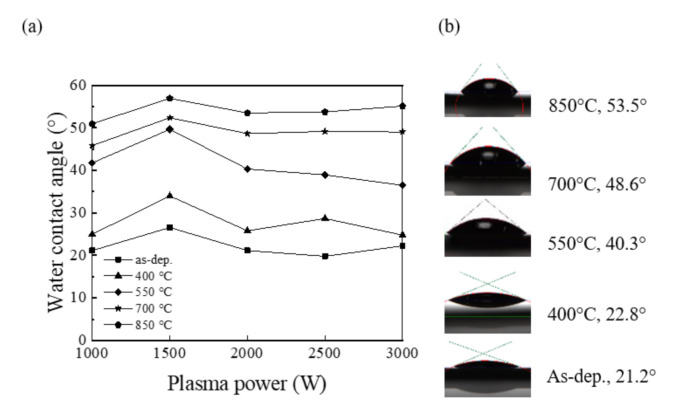
(**a**) WCA for the RP-ALD SiO_2_ films with various plasma powers at different annealing temperatures; (**b**) WCA for the SiO_2_ films when the plasma power is 2000 W at different annealing temperatures.

**Figure 10 nanomaterials-11-01173-f010:**
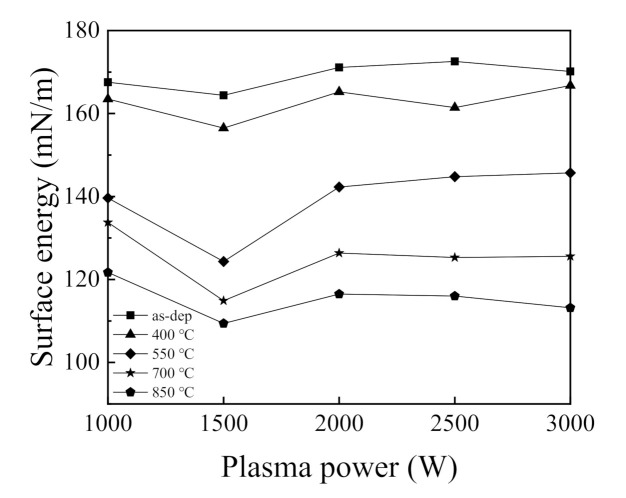
SFE for the RP-ALD SiO_2_ films with various plasma powers at different annealing temperatures.

**Table 1 nanomaterials-11-01173-t001:** Deposition parameters of RP-ALDSiO_2_ thin films.

Parameters	Value
Bubbler temperature (°C)	45
Substrate temperature (°C)	250
BDEAS pulse time (s)	1.6
BDEAS purge time (s)	5
O_2_ pulse time (s)	8
O_2_ flow stabilization (s)	1.4
O_2_ RF power on (s)	6
O_2_ purge time (s)	5
Flow rate of Ar (sccm)	80
Flow rate of O_2_ (sccm)	150
O_2_ plasma power (W)	1000–3000
Flow rate of BDEAS carry gas (sccm)	120
Flow rate of BDEAS dilute gas (sccm)	400

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
