# Peer review of "Deposition and Characterization of RP-ALD SiO2 Thin Films with Different Oxygen Plasma Powers"

_nanomaterials, 2021, doi:10.3390/nano11051173_

Round 1

Reviewer 1 Report

In this paper “Deposition and Characterization of RP-ALD SiO2 Thin Films by Different Oxygen Plasma Power” Authors presents investigations on silicon oxide (SiO2) films were deposited by remote plasma atomic layer deposition with Bis(diethylamino)silane (BDEAS) and oxygen / argon mixture as the precursors. They investigated effects of oxygen plasma powers from 1000 W to 3000 W on the properties of the SiO2 thin films. Authors prepared measurements by atomic force microscope (AFM) and Conductive AFM tests show that the surfaces of the SiO2 thin films with different O2 plasma power are smooth with favourable insulation properties.

In my opinion this paper can be interesting to readers of Nanomaterials journal. The paper is clearly presented. The paper contains 9 figures, 1 table and 1 formula – figures are legible and good quality.

English of the paper is rather good – in my opinion the language of the paper should be a little improved. I am asking for corrections by a native speaker.

I find some mistakes for example:

  • Introduction chapter – In my opinion is too short. Authors should include new information about composites of alumina and graphene-augmented alumina nanofibers. More information based on worldwide (global) study.
  • Experimental Section chapter – Please describe equipment (simulation programs for analysis) used in the theoretical experiment – model of equipment (manufacturer, city, country).
  • Results and Discussion chapter – this chapter should include a discussion related to the topic of the paper (expansion of research methodology, presentation of research results obtained).
  • In my opinion Conclusions chapter should be a little changed. In this chapter there are no summary of all significant information obtained by the Authors and written in the Experimental Section and Results and Discussion
  • Amount of references is also sufficient but some papers cited in the references 14 from all 29 are older then 5 years – these publications constitute over 48 % of all cited papers. I propose to add some new (from the last 5 years) publications on the production or properties of these type of materials. Authors should include several modern papers of global research in this field.
  • Minimum 17 papers from all 29 (over 58 %) are wrote by authors from Asia – China, Japan India and others. Are the topics covered in this article only by research centers from Asia? I ask for a reliable verification of global research in this field. I propose to add some new (from the last 5 years) publications on the production or properties of these type of materials. Authors should include several modern papers (also from Europe and America).
  • In the list of references I found 4 papers of the Authors of reviewed paper – these publications constitute over 13 % of all cited papers – this value is appropriate. But, please indicate the differences in the studies presented in the cited articles and their relation to the presented topic.
  • Please describe all the variables found in the formula (1).
  • Figure 1a – this photography should be as a separate figure with a signature containing a description of all elements of the stand.
  • Figures 2, 3 and 4 should be increased.
  • Figures 6 the quality of CAFM images should be improved.
  • In the whole paper, you write the values in percent as for example 2% (for example Experimental Section chapter) – you should write and value with unit with spaces (2 %).
  • When the Authors mean – "… This work ..." (for example in Funding chapter) the phrase "… this study ..." or "… this investigations ..." should be used, but not the word "… work …".

The results obtained are interesting and promising. The manuscript can be accepted for publication in Nanomaterials journal after MAJOR corrections.

Reviewer 2 Report

The paper "Deposition and Characterization of RP-ALD SiO2 Thin Films by Different Oxygen Plasma Power" by  Xiao-Ying Zhang et al. is very interesting and I recommend its publication in this Journal.
The Introduction is complete and very well describe the contest.
The experimental section is clear and results are well presented, with facts and figures supporting the discussion.
The References are not very recent, I would suggest to include few more recent citations.

Round 2

Reviewer 1 Report

In corrected paper "Deposition and Characterization of RP-ALD SiO2 Thin Films by Different Oxygen Plasma Power" Authors have properly addressed the concerns from the referee. All my remarks have been included in the revised document. Below you will find my comments on the attached answers.

Referring to my substantive reservations – the authors made the necessary modifications. They changed the text of the article and removed stylistic and grammatical errors.

Authors reformatted and extended Introduction, Experimental Section and Results and Discussion chapters. Authors significantly reformat the entire article. Most of the unnecessary figures have been removed – the figures (suggested) has been changed in accordance with the recommendations. They modified a list of a references. Now, paper is preparing in accordance with the journal's guidelines.

The manuscript can be accepted for publication in Nanomaterials journal in the current form.